# Soil Carbon Budget Account for the Sustainability Improvement of a Mediterranean Vineyard Area

**Novara Agata \***, **Favara Valeria, Novara Amelia, Francesca Nicola, Santangelo Tanino, Columba Pietro, Chironi Stefania, Ingrassia Marzia and Gristina Luciano**

Department of Agricultural, Food and Forest Sciences, University of Palermo, Viale Delle Scienze, 13, 90128 Palermo, Italy; favaravaleria@alice.it (F.V.); amelianovara@gmail.com (N.A.); nicola.francesca@unipa.it (F.N.); santangelo.tanino@gmail.com (S.T.); pietro.columba@unipa.it (C.P.); stefania.chironi@unipa.it (C.S.); marzia.ingrassia@unipa.it (I.M.); luciano.gristina@unipa.it (G.L.)

**\*** Correspondence: agata.novara@unipa.it; Tel.: +39-091-2386-22-14

**Abstract:** Sustainable viticulture is suggested as an interesting strategy for achieving the objectives of global greenhouse gas (GHG) emission reduction in terms of mitigation and adaptation. However, knowledge and quantification of the contribution of sustainable vineyard management on climate change impact are needed. Although it is widely assessed by several authors that the agricultural stage has a great impact in the wine chain, very few studies have evaluated the greenhouse gas emission in this phase including the ability of soil to sequester carbon (C) or the off-farm C loss by erosion. This work aimed to provide a vineyard carbon budget (vCB) tool to quantify the impact of grape production on GHG emission including the effects of environmental characteristics and agricultural practices. The vCB was estimated considering four different soil management scenarios: conventional tillage (CT), temporary cover crop with a leguminous species in alternate inter-rows (ACC), temporary cover crop with a leguminous species (CC), permanent cover crop (PCC). The estimation of vCB was applied at territory level in a viticulture area in Sicily (2468 ha of vineyard) using empirical data. Results of the present study showed that the environmental characteristics strongly affect the sustainability of vineyard management; the highest contribution to total $CO_2$ emission is, in fact, given by the C losses by erosion in sloping vineyards. Soils of studied vineyards are a source of $CO_2$ due to the low C inputs and high mineralization rate, except for soil managed by CC which can sequester soil C, contributing positively to vCB. The highest total $CO_2$ emission was estimated in vineyards under CT management (2.31 t ha$^{-1}$y$^{-1}$), followed by CC (1.27 t ha$^{-1}$y$^{-1}$), ACC (0.69 t ha$^{-1}$y$^{-1}$) and PCC (0.64 t ha$^{-1}$y$^{-1}$). Findings of vCB applied at territory level highlighted the key role of the evaluation of carbon budget (CB) on a larger scale to identify the $CO_2$ emission in relation to climatic and environmental factors. The present study could contribute to provide suggestions to policymakers and farmers for reducing GHG emissions and promote more sustainable grape production practices.

**Keywords:** vineyard; carbon budget; soil management; cover crop

## 1. Introduction

Agriculture is one of the economic fields which considerably contributes to global greenhouse gas emissions (GHGs) [1,2]; hence, one of the priorities of 2030 Agenda for Sustainable Development is to guarantee food security impact on climate change. Among crops, viticulture, which covers 4.6 million hectares in the semi-arid Mediterranean basin of the EU [3], could contribute to control the growing GHGs from agriculture sector. The application of more sustainable soil management practices in viticulture systems has proven to control the GHG emission, thanks to the ability of soils

to sequester $CO_2$ as organic matter and plant biomass [4]. A potential measure for mitigation of viticultural GHG emission consists in the adoption of cover crops which has shown to improve soil fertility and microbial activity and, in some environments, consequential increase of topsoil organic matter content [5–7]. In addition, cover crops reduce soil and nutrient erosion, and contribute to preserve the topsoil carbon (C) stock [8].

Several studies demonstrated that emissions in intensive viticulture are mainly due to tillage, fertilization, pest management [9,10]; hence, further mitigation measures must be targeted at inputs reduction (energy use for field management and agro-chemicals). The quantification of $CO_2$ emissions and C sequestration from the viticulture sector could provide useful information to support environmental policies or individual decision making at the farm level for climate change mitigation.

In recent years, several methodologies have been developed by UNFCC (United Nations Framework Convention on Climate Change) and IPCC (Intergovernmental Panel on Climate Change) for GHG calculation. Among the most widespread, the C footprint quantifies the total emissions caused by a product or system across its life cycle [11,12]. The C footprint accurately estimates all emissions of inputs and processes within a specific system boundary. Several studies in the last decade have reported the C footprint of wine [13–16] using calculators like that developed by the International Federation of Wines and Spirts [17]. Most of these calculators do not consider in detail the agronomic phase of the wine production chain. Estimating the agronomic phase of food products presents, in fact, difficulties for data collection of the cycle life inventory due to the inconsistent boundaries, lack of methodology standardization and high data variability for environmental factors and farm management [18,19].

Further efforts should be employed to improve the C budget estimation at the farm level, taking into consideration that soil can became a sink to store $CO_2$ budgets, if sustainable management is applied. Fundamental aspects, related to global C budget such as pruning residue, biomass incorporation, environmental characteristics regulating organic matter mineralization, and C loss, should be introduced in C footprint tools.

The aim of the study was to take a step forward in C footprint estimation of the viticulture sector, providing a vineyard C budget (vCB) tool to quantify the impact of grape production on GHG emission including the effects of environmental characteristics and agricultural practices. The vCB tool will allow for the supply of information to farmers and decision-makers in order to compare and evaluate quantitatively the sustainability of different soil management in vineyard.

## 2. Materials and Methods

### 2.1. Carbon Budget Estimation

The C budget for vineyard management (vCB) was estimated using a calculator tool, which was designed for viticulture farms (Supplementary Material, S1). For the purpose of this work, using the system "from cradle to gate", the carbon budget (CB) was performed only for the vineyard phase, excluding winemaking and wine distribution. The calculator tool follows the approach of life cycle assessment and the framework provided in IPCC for GHGs calculation [20]. The vCB was expressed as Kg $CO_{2eq}ha^{-1}$ of vineyard. The vCB analysis was based on soil and climate characteristics, vineyard and crop residue management, and field energy which include electricity and fuel consumption for agronomic operations (Figure 1). Input data required by the CB calculator are thoroughly described in Table 1. The $CO_2$ emissions from the agriculture phase, which also takes into consideration soil carbon sequestration, microbial mineralization and C loss by erosion, were estimated in relation to environmental characteristics and vineyard management.

**Table 1.** Input data for vineyard carbon budget tool.

| | Parameter | Description | Data Source and Availability |
|---|---|---|---|
| Soil characteristics | Texture (sand, silt, clay) | % | Soil map from regional Sicilian government |
| | Slope and soil length | LS factor (Wishmeier and Smith, 1978) [21] | ESDAC (European Soil Data Centre) https://esdac.jrc.ec.europa.eu/ |
| | Soil pH | | https://esdac.jrc.ec.europa.eu/ |
| | Soil erodibility | K factor (Wishmeier and Smith, 1978) [21] | Panagos et al., 2014 [22] |
| | Soil organic carbon | g kg$^{-1}$ | Soil map from regional Sicilian government |
| | Carbonate | g kg$^{-1}$ | Soil survey |
| | Soil bulk density | t m$^{-3}$ | Soil survey |
| | Cation Exchange Capacity | cmol kg$^{-1}$ | Data from regional Sicilian government |
| | Soil permeability index | From 1 to 6 | Wischmeier et al. (1971) [23] |
| Climate characteristics | Rainfall erosivity | R factor (Mj mm ha$^{-1}$ h$^{-1}$ year$^{-1}$) (Wishmeier and Smith, 1978) [21] | Soil erosion risk, Sicilian Region Fantappiè et al., 2015 [24] |
| | Temperature | Mean annual temperature (°C) | http://www.sias.regione.sicilia.it/ |
| Soil management | Soil tillage | Number and kind of operations | Survey |
| | Fertilization | Fertilizer type, amount of nutrients, number of fertilization | Bouwman et al. (2002) [25] |
| | Pest control | Number of treatments | Survey |
| Crop residue | Cover crop Biomass | t ha$^{-1}$ Mineralization coefficient | Boiffin et al., (1986) [26] |
| | Pruning reside | t ha$^{-1}$ Mineralization coefficient | Fregoni M. (1989) [27] |
| Energy | Fuel Electricity | L h$^{-1}$ horse-power | International Wine Carbon Calculator Protocol—Version 1.2, 2008 [28] |

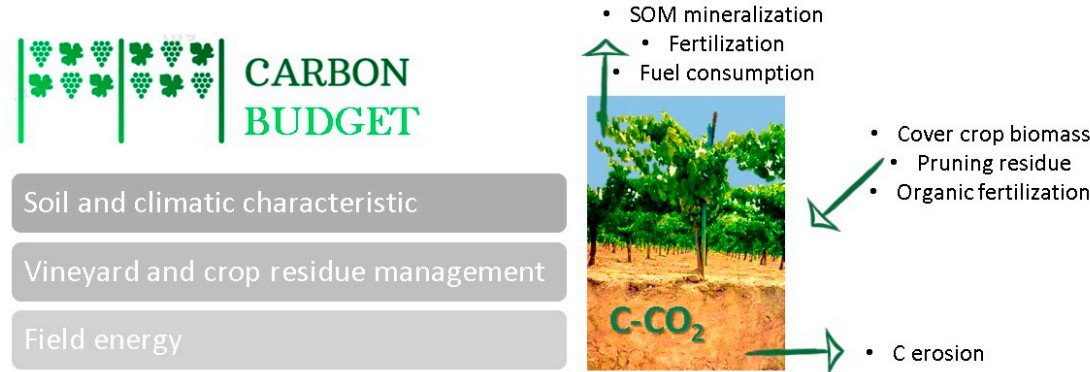

**Figure 1.** Input and output flows considered in the vineyard carbon budget calculator.

The GHG emissions were expressed as Kg of $CO_2$ for the direct emissions or were converted into $CO_2$eq (Kg) for nitrogen emissions. The $CO_2$ emissions due to fossil fuel consumption for agronomic practices were estimated according to the fuel quantity method of the IWCC (International Wine Carbon Calculator) [28].

For the estimations of $NO_x$ emissions due to nitrogen fertilizer, the model of Bouwman et al. [25] was used. Soil physical and chemical properties (e.g., texture, pH, drainage, water retention) and climate factors were used to determine soils NOx emission.

In order to estimate soil C stock change under vineyard, the model proposed by Hénin–Dupuis was adopted [29]. The model simulates the variation of soil organic carbon (SOC) over a long time (vineyard longevity) following exponential kinetic dynamic. In the CB calculator for vineyard management, cover crop biomass and pruning residue were considered the main C inputs. The C budget does not include C stock variation in vine biomass (trunk and root) because it is negligible in comparison to SOC stocks [30], and also because vine plants are removed at the end of their life cycle.

Considering the severity of soil erosion in semiarid vineyards and the consequent C loss by sediment transport [31], the estimation of the off-farm C loss was computed multiplying the SOC content (%) for soil erosion (Mg ha$^{-1}$ y$^{-1}$). Soil erosion was estimated using the USLE equation [21].

*2.2. Data Collection*

The vCB analysis was applied in a viticulture area located in the southern part of Sicily, province of Agrigento. Vineyards belonging to Corbera Winery (2468 ha) were selected (41°64′–41°81′ N and 23°36′–23°62′ E). Climate in the area is typical of a Mediterranean climate, annual precipitation of 520mm, mainly in winter period, and a mean annual temperature of 18°C. (Maximum temperature = 31 °C in July; minimum temperature = 8.8 °C in January). Soils in the study area are loam and clay-loam with a pH range of 6.5–7.4.

The vCB was estimated considering four different soil management scenarios: conventional tillage (CT), temporary cover crop with a leguminous species in alternate inter-rows (ACC), temporary cover crop with a leguminous species (CC), permanent cover crop (PCC). Conventional tillage is still one of the most widespread soil management strategies used by farmers in the area. CT management uses frequent tillage to control weeds and the use of chemical fertilizers and herbicides (Table 2).

**Table 2.** Soil practices for the four different vineyard management scenarios: conventional tillage (CT), cover crop (CC), alternate row cover crop (ACC), permanent cover crop (PCC).

|  | **CT** | **ACC** | **CC** | **PCC** |
|---|---|---|---|---|
| Cover crop | No | Alternate row | Total | Permanent |
| Species | - | Vicia faba | Vicia faba | Trifolium subterraneum |

| Seeding | - | October | October | Each three years |
|---|---|---|---|---|
| Green manure | - | April | April | - |
| Soil cultivation | 6 times | 3 times | 3 times | One time every three years and 2 mowings per year |
| Fertilization | 55 kg N ha⁻¹ | - | - | - |
| Pest control | 3 times | 3 times | 3 times | 3 times |
| Pruning residue management | Buried | Buried | Buried | Not buried |
| Harvesting | Mechanical | Mechanical | Mechanical | Mechanical |

In the last decade, the use of cover crops in vineyards has grown in the selected area. Farmers usually adopt temporary leguminous cover crop, which are seeded in October and buried into the soil in April, in alternate rows, as suggested by best management practices for environmental protection of Sicilian region.

To define vineyard practices, used in these two vineyard managements (CT and ACC), owners of viticulture farms in the selected area were interviewed.

Finally, in order to verify improved management systems in vineyard, the vCB analysis was performed for two further hypothetical scenarios. The CC management includes the use of leguminous cover crop in all rows in order to increase the biomass C input, while PCC management includes a permanent cover crop with a leguminous species characterized by high soil erosion control ability.

Annual practices and inputs for the four different vineyards managements system are described in Table 2. The source of data used in the vCB calculator tool are reported in Table 1. For the estimation of pruning residue biomass data; *Trifolium subterraneum* biomass input = 0.2 t ha⁻¹ recorded in the same study area were used (Pruning residue biomass input = 1.2 t ha⁻¹). For the cover crop biomass, data from previous measurements of dry biomass of *Vicia faba* and *Trifolium subterraneum* carried out in the same environment were used [32]. (*Vicia faba* biomass input = 0.9 t ha⁻¹ for CC and 0.45 t ha⁻¹ for ACC).

## 3. Results and Discussion

### 3.1. Vineyard Carbon Budget

The contribution of SOC dynamics to vCB was relevant and strongly influenced by soil management. Soils, at the same time, act as a sink of $CO_2$ through the increase of SOC stocks or as a source of $CO_2$ when mineralization of SOC is higher than its stabilization. Positive values of soil $CO_2$ emissions indicate a loss of C from soils, on the contrary negative values indicate a net C sequestration (Figure 2).

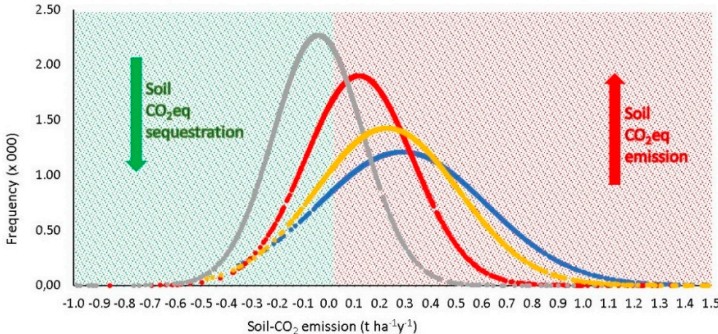

**Figure 2.** Distribution of soil $CO_2$eq emissions (positive values) and soil carbon sequestration (negative values) under different soil management in the selected farms: conventional tillage (blue

line), temporary cover crop with a leguminous species in alternate inter-rows (red line), temporary cover crop with a leguminous species (grey line), permanent cover crop (yellow line).

The main drivers for SOC increase are the C input (cover crop biomass and pruning residue) and soil and climate characteristics which affect the mineralization of soil organic matter. In the case study differences within a selected soil management can be attributed to soil characteristics and initial SOC stocks. Among soil management systems, the different distribution can be ascribed to different C input levels. The lowest value of C input was estimated under CT, where the only C contribution is from pruning residues, followed by PCC, ACC and CC (Figure 2). This result confirmed previous researches which recorded a SOC increase under alternative soil management in comparison to CT, determined by high C input [33,34]. The average value of $CO_2$ emissions from SOC dynamics was 0.30 t $ha^{-1}y^{-1}$ under CT management, followed by PCC (0.23 t $ha^{-1}y^{-1}$) and ACC (0.13 t $ha^{-1}y^{-1}$). A negative average value, which indicates soil C sequestration, was estimated only for the CC scenario (−0.04 t $ha^{-1}y^{-1}$).

The $CO_2$ loss through off-farm C erosion was highest under CT management. Vineyard soils managed by CT are bare almost the whole year and, therefore, high erosion rates are recorded. Cover crops contribute to reduce sediment erosion and consequently C loss, especially under permanent cover crop [32]. The lowest average value of $CO_2$-SOC erosion was under PCC (0.19 ± 0.33 t $ha^{-1}y^{-1}$), followed by ACC (0.44 ± 0.77 t $ha^{-1}y^{-1}$), CC (0.88 ± 1.54 t $ha^{-1}y^{-1}$) and CT (1.37 ± 2.39 t $ha^{-1}y^{-1}$) (Figure 3). The distribution of the soil organic C erosion under PCC has a smaller range in comparison to other management systems, indicating that the use of PCC generally results in negligible C loss by erosion.

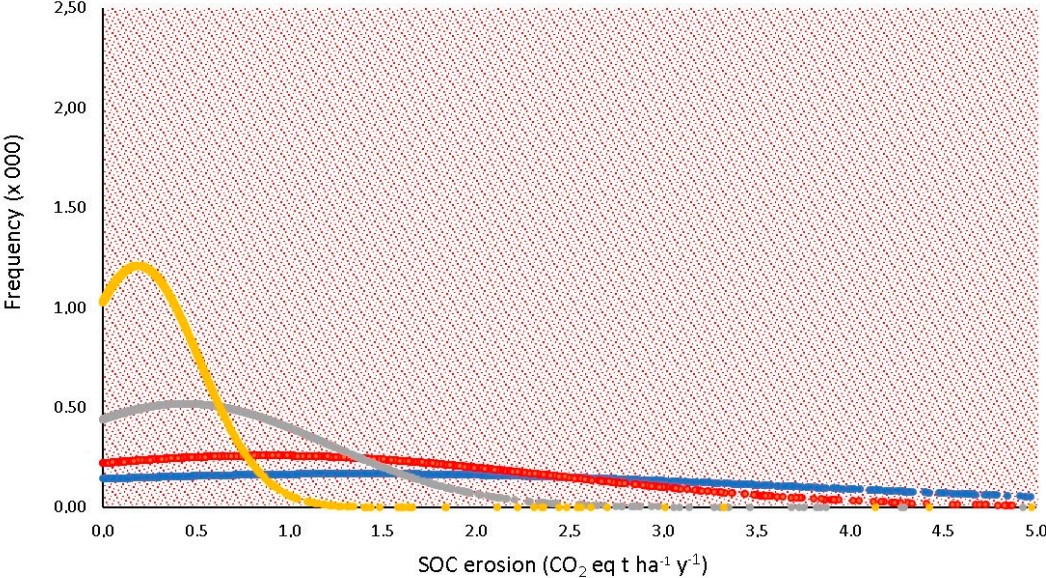

**Figure 3.** Distribution of soil organic carbon erosion ($CO_2$eq t $ha^{-1}$ $y^{-1}$) under different soil management systems: conventional tillage (blue line), temporary cover crop with a leguminous species in alternate inter-rows (red line), temporary cover crop with a leguminous species (grey line), permanent cover crop (yellow line).

In addition to organic matter dynamics (cover crop and residue biomass input), determined by different soil management, and soil C erosion, another source of $CO_2$ emissions is given by agricultural practices (emission from field energy: fuel consumption and electricity) and fertilization. Such emissions are constant for all farms for each management scenario, and not strongly depend on soil or climatic factors. The emissions from nitrogen fertilization, according to the Bouwman et al. [25] model, is affected by soil characteristics (pH, drainage, water retention), which are homogeneous

for the vineyards belonging to Corbera Winery. The emission from soil practices and fertilization were equal to 0.64 t ha$^{-1}$y$^{-1}$ for CT, 0.25 t ha$^{-1}$y$^{-1}$ for CC, 0.28 t ha$^{-1}$y$^{-1}$ for ACC and 0.21 t ha$^{-1}$y$^{-1}$ for PCC. The higher emissions under CT in comparison to the other management systems can be attributed to high number of tillage operation usually used by farmers to control weeds and the high nitrogen fertilization rate. The fertilization rates are, in fact, reduced under cover crop management thanks to the N$_2$ fixation supplied by leguminous species.

The distributions of total CO$_2$ emission for the four different management scenarios are represented in Figure 4. The distribution trends are similar to those of the distribution of SOC erosion, as it is the main source of CO$_2$. The highest total CO$_2$ emission was estimated under CT management (2.31 t ha$^{-1}$y$^{-1}$), followed by CC (1.27 t ha$^{-1}$y$^{-1}$), ACC (0.69 t ha$^{-1}$y$^{-1}$) and PCC (0.64 t ha$^{-1}$y$^{-1}$).

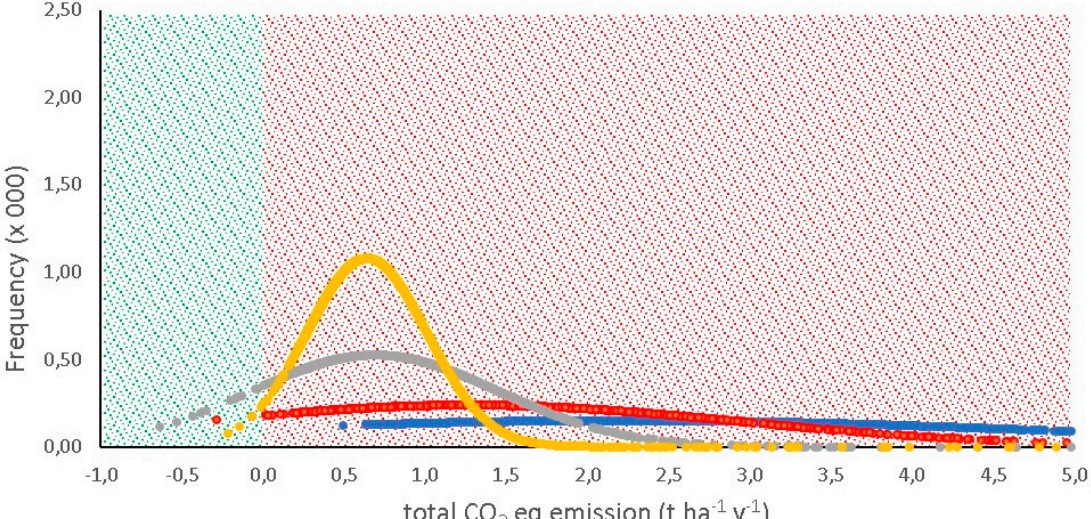

**Figure 4.** Total CO$_2$eq emission (positive values) and carbon sequestration (negative values) under different soil management systems: conventional tillage (blue line), temporary cover crop with a leguminous species in alternate inter-rows (red line), temporary cover crop with a leguminous species (grey line), permanent cover crop (yellow line).

Analyzing the different sources of the whole vCB, the loss of C by erosion is the main CO$_2$ source for CT (60%), CC (70%) and ACC (64%) management (Figure 5). For these three management scenarios, the second source of vCB is the CO$_2$ derived from field energy and fertilization. In PCC management, the main contribution to the whole vCB is from soil emission (37%), followed by field energy and fertilization (33%) and erosion (30%) (Figure 5). The results, shown in Figure 5, allow for isolation of the constrains within each soil management and address strategic environmental measures to reduce CO$_2$ emissions. Comparing different cover crop soil management with CT, the total emissions are reduced by 45% with CC and more than 70% by CC and PCC.

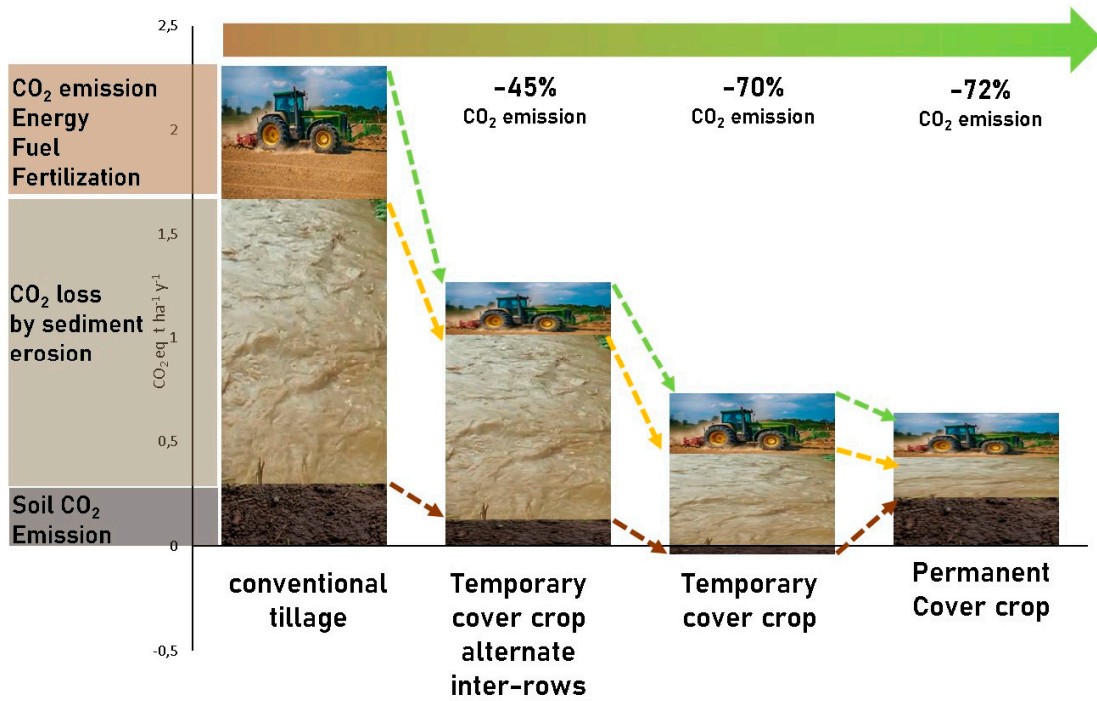

**Figure 5.** Contribution of different sources to vCB (average value of the total area) for the four scenarios.

## 3.2. Scenario Analysis in the Selected Vineyard Area

Considering the selected vineyard area (2468 ha), the trend of $CO_2$ emission among different scenarios and sources reflects the average values of distribution (Figures 6–8). The total carbon emissions under CT amounts to 12391.8 t $CO_2$, followed by ACC (7228 t $CO_2$), CC (3509.2 t $CO_2$) and PCC (3118.5 t $CO_2$) (Table 3). The highest values, recorded under CT, are mainly due to the effect of erosion, which decreases considerably the sustainability of vineyards. The environmental impact of viticulture could be reduced by using cover crops which both decreases the C loss through sediment and increases the biomass C input [8].

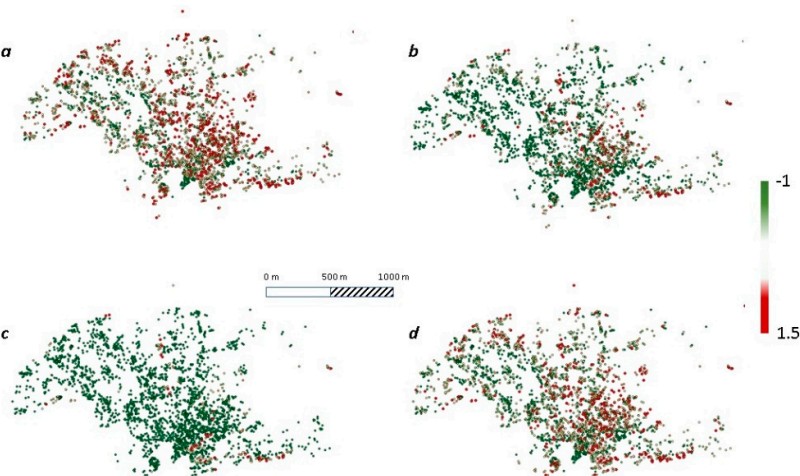

**Figure 6.** Soil CO2eq emissions and soil carbon sequestration under different soil management systems: conventional tillage (**a**), temporary cover crop with a leguminous species in alternate inter-rows (**b**), temporary cover crop with a leguminous species (**c**), permanent cover crop (**d**), for the vineyards of Corbera's winery. Negative values indicate C sequestration.

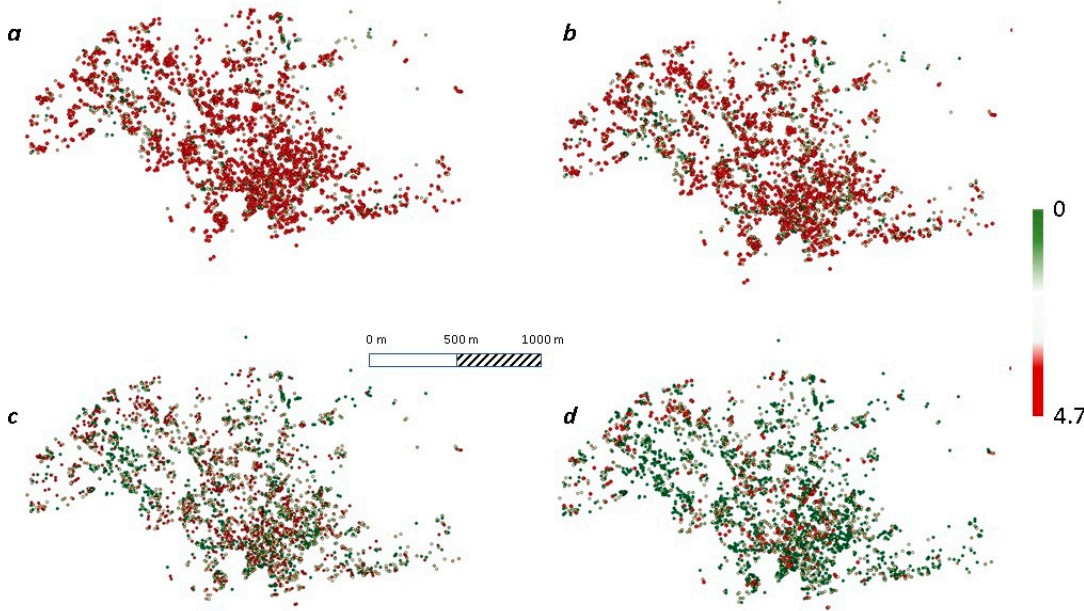

**Figure 7.** Soil organic carbon erosion ($CO_2$eq t ha$^{-1}$ y$^{-1}$) under different soil management systems: conventional tillage (**a**), temporary cover crop with a leguminous species in alternate inter-rows (**b**), temporary cover crop with a leguminous species (**c**), permanent cover crop (**d**) for the vineyards of Corbera's winery.

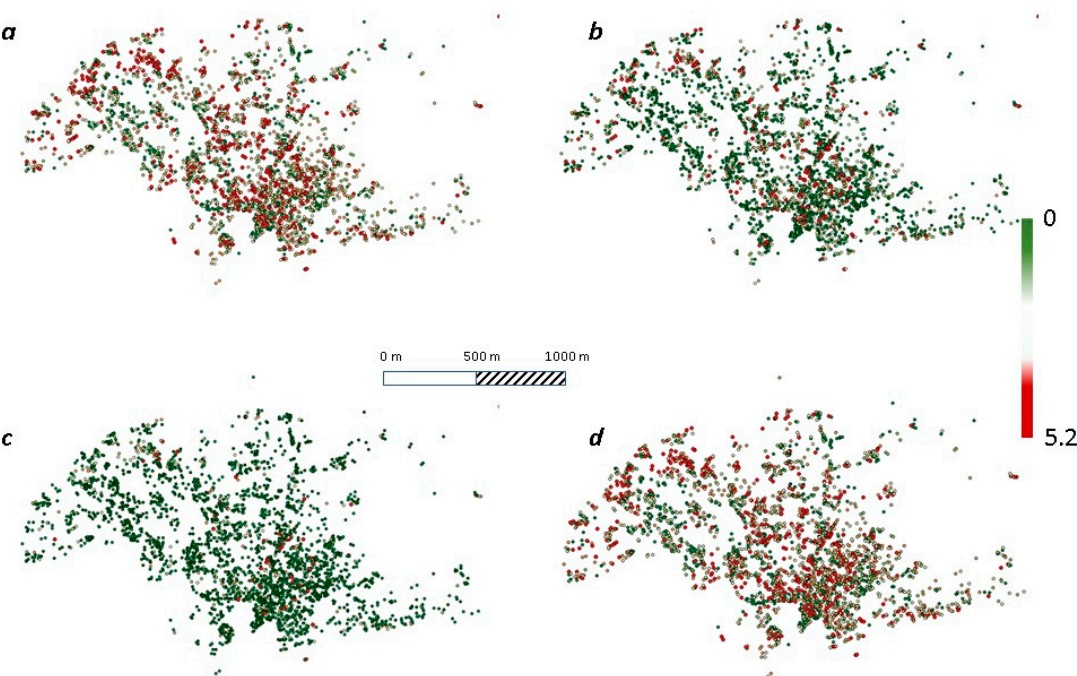

**Figure 8.** Total $CO_2$eq emission and carbon sequestration under different soil management systems: conventional tillage (**a**), temporary cover crop with a leguminous species in alternate inter-rows (**b**), temporary cover crop with a leguminous species (**c**), permanent cover crop (**d**) for the vineyards of Corbera's winery.

**Table 3.** $CO_2$ emissions from different categories and total carbon budget (t $CO_2$) from 2468 ha (Corbera vineyards) for different soil management scenarios.

| CO₂eq Emission | CT | ACC | CC | PCC |
|---|---|---|---|---|
| Soil emission | 1615.7 | 707.6 | −200.6 | 1279.4 |
| Soil carbon erosion | 8995.3 | 5812.4 | 2906.2 | 1245.5 |
| Field energy fertilization | 1780.7 | 708.4 | 803.6 | 593.6 |
| Total carbon budget | 12391.8 | 7228.3 | 3509.2 | 3118.5 |

The PCC scenario shows a higher soil $CO_2$ emission in comparison to CC and ACC, because of the low annual C input of *Trifolium subterraneum* and because the pruning residues are not buried.

The analysis of vCB obtained from the whole vineyard area allows for the evaluation of sources that have a fundamental role in the contribution of $CO_2$ emission. In order to improve the sustainability of viticulture in the studied area, on-farm tracking, pinpointing sources of emissions and adopting agri-environmental strategies are needed, especially for the farms with highest vCB.

## 4. Implication and Conclusion

The present work contributes to improving the sustainability of vineyard management in Mediterranean environments through the comparison and the quantification of the impact in terms of $CO_2$ emissions of different agronomic practices. Although the effect of best management practices on soil fertility and ecosystem service benefits are well-known to the academic community, their global quantification is often difficult due to the high environmental variability [7]. It is believed that a quantification of viticulture impacts could be helpful to increase the awareness of policy makers and farmer perceptions of environmental risks.

The vCB tool integrates and considers some aspects which are not present in the most common carbon footprint calculators, introducing the variation of SOC stock and the loss of C due to erosion processes. Findings of this work confirmed the role of soil for C sequestration, following best management practices, such as the use of temporary CC in all vineyard inter-rows. The portion of C sequestered is relevant and, therefore, CC should be promoted and further investigated because it contributes to the reduction of $CO_2$ emissions of the whole wine production chain.

Moreover, results highlighted the severity of erosion in Mediterranean environments on the total CB. In sloping vineyards, the main source of C loss is due to erosion and therefore the use of cover crop should be mandatory especially in the rainiest season.

The small amount of required data allows for the application of the vCB tool at territory, as results show. This application highlights the environmental variability of $CO_2$ emission, considering a constant soil management; hence, it could be helpful to modulate vineyard management protocols or incentives in relation to farm environmental characteristics.

In addition, the vCB tool, accessible and handy for farmers, is useful for a quantitative self-evaluation in order to identify the weak points of their grape production. For wine producers, the quantification of vineyard $CO_2$ emission is an opportunity to demonstrate the production sustainability to consumers who are interested in organic products, not only for human health, but also for decreasing the environment impact.

**Supplementary Materials:** The following are available online at www.mdpi.com/xxx/s1, S1 vineyard carbon budget tool.

**Author Contributions:** Conceptualization, A.N. and L.G.; data curation, A.N. and L.G.; funding acquisition, N.F.; investigation, L.G.; methodology, V.F., A.N. and T.S.; project administration, N.F.; supervision, L.G.; writing—original draft, A.N.; writing—review and editing, P.C., S.C. and M.I. All authors have read and agreed to the published version of the manuscript.

**Funding:** This research was funded by 2017-NAZ-0228—CUP B78117000260008, Ministero dello Sviluppo Economico, grant number F/050267/03/X32 PON lmprese and Competitività 2014–2020, Asse 1–Azione 1.1.3, Programmazione Europea Horizon2020.

**Conflicts of Interest:** The authors declare no conflicts of interest. The funders had no role in the design of the study; in the collection, analyses, or interpretation of data; in the writing of the manuscript, or in the decision to publish the results.

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
