# Peer review of "Soil Carbon Budget Account for the Sustainability Improvement of a Mediterranean Vineyard Area"

_agronomy, doi:10.3390/agronomy10030336_

Round 1

Reviewer 1 Report

Overall this paper is very well-written, easy to follow, the results are meaningful and the conclusions are well-supported by the data presented.  The changes to be made are minor:

check the use of superscripts and subscripts throughout the manuscript, make the suggested English language corrections (see attached PDF), perhaps define the term "field energy" or choose a different term -- the meaning is not clear to me, incorporate the short paragraph in lines 48-50 into the previous paragraph, incorporate the short paragraph in lines 51-53 into the following paragraph, improve the clarity of the key message in lines 118-126 (see suggestion in attached PDF), remove the reference to chapter 3.1 in line 207, change the numbers on the scale in figures 6, 7, 8 (see note next to figure 6 in the attached PDF), format Appendix A/1 as supplementary material fill out the sections on supplementary materials, author contributions, acknowledgements, conflicts of interest, incorporate references in lines 352-364 into the reference list.

Author Response

Overall this paper is very well-written, easy to follow, the results are meaningful and the conclusions are well-supported by the data presented.  The changes to be made are minor:

Point 1 check the use of superscripts and subscripts throughout the manuscript,

Response 1: (checked)

Point 2: make the suggested English language corrections (see attached PDF),

Response 2: (thanks, done)

Point 3: perhaps define the term "field energy" or choose a different term -- the meaning is not clear to me, Response 3: The term field energy indicates the consumption of energy for irrigation, pruning, tillage, pesticide and fertilizer application and manure transportation, and electricity. This term is widely applied in Carbon footprint studies, see for instance V.D. Litskas et al. / Journal of Cleaner Production 156 (2017) 418e425.; or  O Svubure et al.,  Carbon footprinting of potato (Solanum tuberosum L.) production systems in Zimbabwe.

We added a sentence to explain the meaning of field energy in the line 87.

Point 4 incorporate the short paragraph in lines 48-50 into the previous paragraph, incorporate the short paragraph in lines 51-53 into the following paragraph

Response4: (done)

Point 5: improve the clarity of the key message in lines 118-126 (see suggestion in attached PDF),

Response 5: This part was modified.

Point 6: remove the reference to chapter 3.1 in line 207,

Response6: (done)

Point 7: change the numbers on the scale in figures 6, 7, 8 (see note next to figure 6 in the attached PDF), Response 7: (done)

Point 8: format Appendix A/1 as supplementary material fill out the sections on supplementary materials, author contributions, acknowledgements, conflicts of interest, incorporate references in lines 352-364 into the reference list.

Response 8: Ok

Reviewer 2 Report

GENERAL COMMENTS

The manuscript raises some interesting points with respect to carbon sources and sinks in vineyard ecosystems. The model outputs also emphasise the potential importance of soil organic carbon and carbon loss processes (such as erosion or decomposition/respiration) in modulating soil/ecosystem carbon storage or loss. The approach the authors have used has merits and clearly addresses some of the knowledge gaps they identified in their introduction. However, my major concern with this paper is that the authors have not adequately parameterised and tested the C balance model they have used to generate these data. Best-practice would be to parameterise the model using  a sub-set of empirical data and to also set aside a separate sub-set of data for accuracy testing (e.g. using plots of observed versus predicted values to determine how accurately the model simulates reality; using bootstrapping to test model consistency, etc.). However, in this study, the authors have not collected any empirical field data for model parameterisation and testing, which means that the results generated from this modelling exercise are highly speculative and are not sufficiently grounded in reality for us to have a high degree of confidence in the findings. While this paper represents a promising start, but the lack of empirical grounding means that we have no certainty or confidence in the model results.

SPECIFIC COMMENTS

Lines 17-22: The authors need to clarify what methodology they have used in this section of the abstract, as it is not clear whether they have used empirical measurements, modelling or some combination of the two approaches. Lines 70-74: The authors clearly outline the current knowledge gaps (lines 54-69) for C budgeting and C footprint analysis, but need to be more specific about what measurements go into their vCB which represent an improvement over current methods. I recommend expanding this section slightly to provide the reader with more information on what precisely they measure in vCB which makes it so much better than the current approach. For example, do they measure the C fluxes identified in lines 65-69 (i.e. pruning residue, biomass incorporation, C losses from respiration and erosion, etc.)? Methods (Lines 77-132): This section describes the C budget approach (lines 77-104) and the study sites (lines 105-132) for which C budgets were constructed. However, what is problematic here is that the authors do not appear to have collected empirical data from the study sites in order to test and parameterise their C budget model, nor have they conducted a sensitivity analysis (using the site-specific data) to assess how the model responds to different driving variables. This concerns me because the authors have not tested the accuracy of their modelled data against empirical observations, nor are they able to test the consistency of the model or range of variance (which would be possible using something like bootstraping). Lines 149-157: Studies of soil and crop management in other agri-ecosystems have found similar patterns, i.e. lower soil C storage in conventional tillage systems, higher soil C storage in systems where soil C inputs are elevated. However, the authors do not reference these other studies, even though they are pertinent in understanding the trends identified here.

Author Response

Reviewer 2

Point 1. The manuscript raises some interesting points with respect to carbon sources and sinks in vineyard ecosystems. The model outputs also emphasise the potential importance of soil organic carbon and carbon loss processes (such as erosion or decomposition/respiration) in modulating soil/ecosystem carbon storage or loss. The approach the authors have used has merits and clearly addresses some of the knowledge gaps they identified in their introduction. However, my major concern with this paper is that the authors have not adequately parameterised and tested the C balance model they have used to generate these data. Best-practice would be to parameterise the model using a sub-set of empirical data and to also set aside a separate sub-set of data for accuracy testing (e.g. using plots of observed versus predicted values to determine how accurately the model simulates reality; using bootstrapping to test model consistency, etc.). However, in this study, the authors have not collected any empirical field data for model parameterisation and testing, which means that the results generated from this modelling exercise are highly speculative and are not sufficiently grounded in reality for us to have a high degree of confidence in the findings. While this paper represents a promising start, but the lack of empirical grounding means that we have no certainty or confidence in the model results.

Response 1. The carbon footprint analysis, as well as the vCB tool, is not an empirical model. It is an estimation of all sources of CO2 emission of a product chain and therefore it cannot be considered a model. The calculation through vCB uses previous equation such as USLE for erosion estimation, or the equation of Bowman for NOx emission, or the IWCC for fuel emission (see appendix A).

We don’t agree with reviewer and we guess that it is not necessary tested again these equations which are widely assessed by scientific community and used by several researchers over the last years. 

All data used in our calculator were collected. The source of data is described in appendix a

SPECIFIC COMMENTS

Point 2. Lines 17-22: The authors need to clarify what methodology they have used in this section of the abstract, as it is not clear whether they have used empirical measurements, modelling or some combination of the two approaches.

Response 2. In the abstract it was added that empirical data were used)

Point 3. Lines 70-74: The authors clearly outline the current knowledge gaps (lines 54-69) for C budgeting and C footprint analysis, but need to be more specific about what measurements go into their vCB which represent an improvement over current methods. I recommend expanding this section slightly to provide the reader with more information on what precisely they measure in vCB which makes it so much better than the current approach. For example, do they measure the C fluxes identified in lines 65-69 (i.e. pruning residue, biomass incorporation, C losses from respiration and erosion, etc.)?

Response 3. The current knowledge gaps are well defined in line 65-69; the vCB improves the current methods, adding in the carbon budget the calculation of CO2 emission or C sequestration from pruning residue, biomass incorporation, C losses from respiration and erosion. Therefore, it is helpful to provide information on the impact of a specific soil management system. Calculation are all derived from specific database, reported in Appendix A and specifically from Sicilian region maps, ESDAC jrc etc. For other specific data concerning the single farm in the considered area, Corbera winery database was used.)

Point 4. Methods (Lines 77-132): This section describes the C budget approach (lines 77-104) and the study sites (lines 105-132) for which C budgets were constructed.  However, what is problematic here is that the authors do not appear to have collected empirical data from the study sites in order to test and parameterise their C budget model, nor have they conducted a sensitivity analysis (using the site-specific data) to assess how the model responds to different driving variables. This concerns me because the authors have not tested the accuracy of their modelled data against empirical observations, nor are they able to test the consistency of the model or range of variance (which would be possible using something like bootstraping).

Response 4. Regarding data sources, see the previous answer and the Appendix A.  The carbon budget account is not a model, it uses previous model/equation, widely tested

Point 5 Lines 149-157: Studies of soil and crop management in other agri-ecosystems have found similar patterns, i.e. lower soil C storage in conventional tillage systems, higher soil C storage in systems where soil C inputs are elevated. However, the authors do not reference these other studies, even though they are pertinent in understanding the trends identified here.

Answer 5 The comparison of our results with other studies was added. References were added

Reviewer 3 Report

The topic looks interesting and focussing on the current issue. The authors need improve the methodology and separately  discuss the results and discussions to deliver elaborately his/her message.

The author needs to elaborate about the tools showing some imperical equations if appropriate.

Better to separate results and discussions-where the author need to discuss the best options among the management and the reason. How (or % contribution) this management contribute in GHG mitigations nationally.

Some points need to be concentrate like space between two words, acronyms, superscript/subscript-I mentioned some-but not all.

In abstract full acronym: For instance GHG in line 12

L19: space

L24: subscript CO2

L28: superscript 2.31 t ha-1y-1),

Line 42, line 181:  thanks to the…. Thanks in the paper OK?/formal? Can you rephrase the significance?

C sometimes as C ( L 51)and carbon (e.g in abstract several lines)

L68: Comma after C loss?

L81: wine making-word together?

L91: in Nitrogen not in small n?

Should nitrogen and carbon written as full always or details in first instance and then use acronym?

L113: CT or conventional tillage? Please see previous line.

L140: CO2 emission (space)

L199: Check the line if correct.

L246: space

L256: what is meant impact of CB?

L289: soil bulk density unit?

Regarding weather information: please provide the maximum temperature and rainfall and the month when it occurs.

In addition, I think need to input soil properties like soil texture, pH etc. The link in appendix might not enough.

L153: please mention the Fig for annual C input under each management.

L172: better to specify the emissions (CO2?)

Figure 6,7,8: The number indicates in the bar-what is about positive and negative values?

33-24: different font?

Author Response

Point 1. The topic looks interesting and focussing on the current issue. The authors need improve the methodology and separately discuss the results and discussions to deliver elaborately his/her message.

Answer 1. In order to avoid a boring result section, we prefer to merge result and discussion section

Point 2. The author needs to elaborate about the tools showing some imperical equations if appropriate.

Answer 2. The equations are in the last page of the tool.

Point 3. Better to separate results and discussions-where the author need to discuss the best options among the management and the reason.

Answer3 see answer 1.

Point 4. Some points need to be concentrate like space between two words, acronyms, superscript/subscript-I mentioned some-but not all.

Answer4  see answer 1

Point 5. In abstract full acronym: For instance GHG in line 12

Answer5.  It was changed

Point 6 L19: spaceL24: subscript CO2 (done) L28: superscript 2.31 t ha-1y-1)

Answer 6. Corrected

Point 7 Line 42, line 181:  thanks to the…. Thanks in the paper OK?/formal? Can you rephrase the significance?

Answer 7. The English was checked

Point 8 C sometimes as C ( L 51) and carbon (e.g in abstract several lines)

Answer 8. Corrected

Point 9  L68: Comma after C loss?  L81: wine making-word together?Yes  L91: in Nitrogen not in small n? ok Should nitrogen and carbon written as full always or details in first instance and then use acronym? Ok corrected L113: CT or conventional tillage? Please see previous line. Ok  L140: CO2 emission (space) ok

Answer 9 All suggestion were accepted

Point 10. L199: Check the line if correct.

Answer 10. It was corrected

Point 11 L246: space

Answer 11. Corrected

Point 12. L256: what is meant impact of CB?

Answer 12. The sentence was modified

Point 13. L289: soil bulk density unit?

Answer 13. The unit was modified

Point 14. Regarding weather information: please provide the maximum temperature and rainfall and the month when it occurs. In addition, I think need to input soil properties like soil texture, pH etc. The link in appendix might not enough.

Answer 14. A sentence with these details was added in mat and met.

Point 15 L153: please mention the Fig for annual C input under each management.

Answer 15. I’m sorry, but the question is not clear for us.

Point 16. L172: better to specify the emissions (CO2?)

Answer 16. Corrected.

Point 17. Figure 6,7,8: The number indicates in the bar-what is about positive and negative values? In the figure 6 it was added the meaning of negative values.

Answer 17. Corrected

Point 18. 33-24: different font? Ok

Answer 18. Corrected

Round 2

Reviewer 3 Report

I am not completely agreeing with response of point 1. There are several scientific papers describing nicely results and discussion part separately which is NOT boring.

Point 4: Can you check your answer regarding point 4?

Point 5: Sorry I could not see the changes.

L254-255: because twice, Trifolium subterraneum-should not be italic?

Point 13: please check.

Point 15: Sorry not to make clear. You can provide annual plant C input values from different sources e.g. pruning, cover crop, annual C loss from erosion, respiration under three scenarios.

Author Response

We thanks the reviewer for the helpful contribution to improve our manuscript and we agree with all his comments. Our answer to the specific comments are reported below. 

Point 4: Can you check your answer regarding point 4? 

Answer4: Sorry for this misunderstanding. We have checked again the editing of the manuscript.

Point 5: Sorry I could not see the changes.

Answer 5. The abstract was changed according to reviewer’s comments

L254-255: because twice, Trifolium subterraneum-should not be italic?

Answer: According to author guidelines, the botanical name must be written in italic.

Point 13: please check.

Answer 13 : The unit of measurement of soil bulk density is Mg m-3 or t ha-1 . We have changed with t ha-1

Point 15: Sorry not to make clear. You can provide annual plant C input values from different sources e.g. pruning, cover crop, annual C loss from erosion, respiration under three scenarios.

Answer 15: In material and method, the annual carbon input by cover crop biomass and pruning residue were added.

The annual C loss from erosion is represented in figure 3. The values are reported in CO2 to be comparable with other results

The respiration (soil component) is represented in figure 2.